# Identifying and overcoming barriers to automated external defibrillator use by GoodSAM volunteer first responders in out-of-hospital cardiac arrest using the Theoretical Domains Framework and Behaviour Change Wheel: a qualitative study

Christopher M Smith [1], Frances Griffiths [2], Rachael T Fothergill [3], Ivo Vlaev,[4] Gavin D Perkins[1]

For numbered affiliations see end of article.

**Correspondence to**
Dr Christopher M Smith; c.smith.20@warwick.ac.uk

## ABSTRACT

**Objectives** GoodSAM is a mobile phone app that integrates with UK ambulance services. During a 999 call, if a call handler diagnoses cardiac arrest, nearby volunteer first responders registered with the app are alerted. They can give cardiopulmonary resuscitation (CPR) and/or use a public access automated external defibrillator (AED). We aimed to identify means of increasing AED use by GoodSAM first responders.

**Methods** We conducted semistructured telephone interviews, using the Theoretical Domains Framework to identify and classify barriers to AED use. We analysed findings using the Capability, Opportunity, Motivation, Behaviour (COM-B) model and subsequently used the Behaviour Change Wheel to develop potential interventions to improve AED use.

**Setting** London, UK.

**Participants** GoodSAM first responders alerted in the previous 7 days about a cardiac arrest.

**Results** We conducted 30 telephone interviews in two batches in July and October 2018. A public access AED was taken to scene once, one had already been attached on scene another time and three participants took their own AEDs when responding. Most first responders felt capable and motivated to use public access AEDs but were concerned about delaying CPR if they retrieved one and frustrated when arriving after the ambulance service. They perceived lack of opportunities due to unavailable and inaccessible AEDs, particularly out of hours. We subsequently developed 13 potential interventions to increase AED use for future testing.

**Conclusions** GoodSAM first responders used AEDs occasionally, despite a capability and motivation to do so. Those operating volunteer first responder systems should consider our proposed interventions to improve AED use. Of particular clinical importance are: highlighting AED location and providing route/time estimates to the patient via the nearest AED. This would help single responders make appropriate decisions about AED retrieval. As AED

### Strengths and limitations of this study

► This interview study is the first time behaviours affecting automated external defibrillator (AED) use of volunteer first responders activated to out-of-hospital cardiac arrest have been explored using the Theoretical Domains Framework and Behaviour Change Wheel.

► We have developed a list of potential interventions to increase AED use that can be tested by any similar first responder system, of which there are a number worldwide.

► Actual AED use in this study was infrequent: interview responses focus on intended actions and may not be representative of participants' behaviours in future time-critical emergency situations.

collection may extend time to reach the patient, where there is sufficient density of potential responders, systems could send one responder to initiate CPR and another to collect an AED.

## INTRODUCTION

Fewer than 1 in 10 patients who suffer an out-of-hospital cardiac arrest (OHCA) survive to hospital discharge in England.[1] Public access defibrillation (PAD)—bystander use of public access automated external defibrillators (AEDs)—is associated with doubled survival to hospital discharge with good neurological function.[2] However, PAD is used in fewer than 5% of OHCA.[1 3]

### Volunteer first responder systems

Mobile app-based volunteer first responder systems are one approach to overcome

limited PAD use.[4] A number of countries have app-based or text message systems to alert bystanders to a nearby OHCA, including Netherlands,[5] Sweden[6] and USA.[7] 'GoodSAM' (https://www.goodsamapp.org/) is the system that operates in the UK.[8] Patient outcome data are limited.[9] The only randomised controlled trial, comparing a text message alert system supplementing ambulance service care versus ambulance service care alone, reported significantly higher bystander cardiopulmonary resuscitation (CPR) rates (primary outcome: 61.6% vs 47.8%, p<0.001) but no significant difference in the secondary outcome of 30-day survival (11.2% vs 8.6%, p=0.28). However, there was no mention of PAD in this system.[6] In an observational study in the Netherlands (2012–2014), OHCA patients attended by text message-alerted first responders were 2.8 times more likely to survive to hospital discharge than those for whom alerted first responders did not attend. First responders attached an AED in 26.8% cases.[5]

## Barriers and facilitators to PAD

A systematic review considering barriers and facilitators to PAD reported that public access AEDs are located close to few OHCA, and their accessibility is reduced out of hours. Knowledge, awareness and willingness to use public access AED vary but are increased by prior CPR/AED training.[3] In volunteer first responder systems, not reaching the scene after accepting an alert,[5 7 10] not being the first to arrive and perform CPR[7 11] or attach an AED[10] have all been reported.

There is limited knowledge about the effect of first responder systems and how to most effectively deploy AEDs within them and recognition that we need high-quality research to investigate this.[4] The Scandinavian AED and Mobile Bystander Activation trial[12] is currently randomising volunteer first responders using the 'Heart-runner' app in Denmark and Sweden to either: all responders travel direct to the patient or some responders retrieve the nearest AED first. The primary outcome is the proportion of patients who have an AED attached before the ambulance service arrives.

## Theoretical frameworks to develop interventions for increased PAD use

There is no work using a theoretically informed behavioural framework to address low AED use in volunteer first responder systems. The Theoretical Domains Framework (TDF) can be used to identify specific barriers or facilitators to a particular outcome.[13 14] The 14 domains of the revised version of the TDF[15] can be further grouped and integrated with the Capability, Opportunity, Motivation, Behaviour (COM-B) framework. COM-B characterises three targets for behavioural change in order to inform the design of healthcare interventions and can, in turn, be linked to the Behaviour Change Wheel (BCW).[16] The BCW is used to develop interventions and the means of implementing them.

## Research question

How can we increase the numbers of AEDs taken to an OHCA by a GoodSAM first responder following an alert?

To answer this, we took data from an interview study and used the BCW to develop a theoretically informed list of interventions to increase public access AED use by GoodSAM first responders during an alert.

## METHODS
## Description of GoodSAM system

GoodSAM is an app-based system that integrates with several local ambulance services to alert trained volunteers to an OHCA.[8] In the UK, GoodSAM first responders require at least in-date CPR training to register with the app. A call handler, using a computer-aided dispatch system during a 999 call, allocates a code identifying the clinical problem. If this code indicates that a patient is suffering a current or imminent cardiac arrest, the system automatically activates GoodSAM. First responders can accept or decline this alert. The app displays incident address, route and nearby public access AEDs. They decide themselves whether to retrieve an AED or travel directly to scene: they receive no instruction via the app.

There were 298 OHCAs in London in July 2018 and 306 in October 2018.[17] There are 3–5 GoodSAM notifications daily for OHCA. In July 2018, GoodSAM responders accepted 19% of alerts in London (information from GoodSAM).

## Study setting

London, UK. GoodSAM integrated with London Ambulance Service in October 2015. Following an OHCA diagnosis during the 999 call, GoodSAM first responders within 300 m of the incident receive an alert siren on their mobile device.

## Study design

Telephone semistructured interviews performed with GoodSAM first responders in London, with responses coded to TDF domains. We developed topic guides based on COM-B. The focus in first responder interviews was on decisions to use a public access AED.

## Participant selection and recruitment

We approached first responders consecutively in two blocks in July and October 2018, shortly after they received an alert in London. We made an a priori decision to conduct 30 interviews before assessing for data saturation—defined here as no new information emerging from interviews 28–30.[18] We felt this was pragmatic and consistent with other studies using qualitative interviews.[19]

## Interview procedure

We approached potential participants via email, with a participant information sheet and consent statements attached. Interested participants replied to CMS, who subsequently arranged an interview. We sent no further emails to non-respondents.

CMS conducted interviews with GoodSAM first responders using Microsoft Skype (audio only) to the telephone number provided and recorded directly onto computer using QuickTime player. CMS introduced himself as a clinician but explained his current role as a university researcher undertaking a PhD. Interviewer and participants did not previously know one another. Participants could ask questions before recording started. Recording began and participants gave verbal agreement to consent statements, which the interviewer read aloud. Interviews lasted a median of 14:56 min (range 7:41–24:01) and recording continued until the call ended. All interviews took place at least 24 hours after the initial invitation and within 7 days of the GoodSAM alert.

We kept audio files and transcriptions in separate encrypted computer folders. No participant-identifying information was associated or stored with audio files or transcripts.

### Interview data analysis

CMS transcribed and coded interviews using NVivo (QSR International). CMS and FG subjected the first three interviews to detailed line-by-line coding without reference to the TDF: they judged that the codes and emerging themes sufficiently matched with TDF domains for us to comprehensively code information to these TDF domains. CMS subsequently coded all interviews using the TDF, with periodic checking undertaken by FG and iterative update and review of coding as required. We remained alert to data that did not fit into a TDF domain. We matched coded material from TDF domains to components of COM-B, with which the TDF integrates.[16] We subsequently analysed the data and synthesised a narrative about the first responders' experiences relating to their capability (subdivided into physical and psychological), opportunity (social and physical) and motivation (automatic and reflective) to use an AED as part of their response.

### Developing interventions

We developed potential interventions to improve AED use by GoodSAM first responders, using a step-by-step approach described in the BCW. As described,[16] we grouped and integrated TDF domains into the COM-B framework which, in turn, were linked to the BCW. Using the BCW, we identified relevant intervention functions for behavioural determinants that would have to change to increase AED use. We further assessed these intervention functions (from a list of nine in the BCW) using affordability, practicability, effectiveness/cost-effectiveness, safety and efficacy criteria and identified policy categories, behavioural change techniques and modes of delivery. We selected behavioural change techniques described in the Behaviour Change Techniques Taxonomy version 1[20]: these associate with intervention functions in the BCW.[16]

Ultimately, we decided that presenting our findings—initially coded to TDF domains—according to whether they represented capability, opportunity or motivation to change behaviour, was the most accessible way to communicate our findings.

### Patient and public involvement

There were two lay representatives on the project's advisory committee. They contributed to the topic guide, participant information sheets, consent forms and protocol. Patients and/or the public were involved in the design, or conduct, or reporting, or dissemination plans of this research.

### Reporting

We report this study according to Standards for Reporting Qualitative Research[21] (online supplementary file).

## RESULTS

### Interview study

We conducted 30 telephone interviews, from 248 email invitations. Twenty-one interviews took place in July 2018. After reviewing the topic guide, we conducted the remaining interviews in October 2018. A higher proportion of our first 21 participants had accepted their latest alert (12/21, 57%) compared with the London average, so for these last nine interviews, we invited only those who had declined their latest alert. We determined that we had achieved data saturation after 30 interviews.

Ultimately, 14/30 (47%) responders accepted their most recent alert. Six reached the patient's side; three before the ambulance service. Four of 11 patients, where known, were in cardiac arrest. Most participants had also received previous alerts so we also discussed issues relating to these alerts, starting with the most recent.

Eleven first responders had previous healthcare experience. Overall figures for all GoodSAM first responders were not available.

We present a summary of barriers and facilitators to AED use in figures 1–3.

### Physical capability

No participant said they felt unable to provide CPR or use an AED, and all who commented believed previous CPR/AED training facilitated CPR and AED use when

| Facilitators |
| --- |
| Previous training in CPR/AED use |
| Previous real-life experience in CPR/AED use |
| Good awareness of Public Access Defibrillation |
| Using the app to check AED locations before an alert |
| Most participants felt capable and competent to respond to alerts |

| Barriers |
| --- |
| Being less familiar of AED locations in unfamiliar areas |
| Varying recall of information given during an actual alert |
| Not recalling whether or not AED location were displayed at time of the alert |
| Not even considering AED retrieval at the time of the alert |

**Figure 1** Possible barriers and facilitators to AED use: capability. AED, automated external defibrillator.

| Facilitators |
| --- |
| Having one's own AED (Some first-responders did and had taken them to the scene) |
| Previously known AED available nearby |
| AED already present on scene |

| Barriers |
| --- |
| Public-access AEDs |
| Perceived to be too far away from first-responder or patient at time of alert |
| Perceived difficulty finding exact location |
| Less availability during out-of-hours alerts |
| Perceived to be inaccessible in locked cabinets |
| App-specific issues: |
| GoodSAM location inaccurate at time of alert |
| Alert siren not always heard |
| App slow to display incident location once alert accepted |
| Healthcare professionals unable to leave patients at work to respond |
| Non-healthcare professionals feeling unable to leave work or dependents |
| Arriving after the ambulance service |

**Figure 2** Possible barriers and facilitators to AED use: opportunity. AED, automated external defibrillator.

responding. Eight participants reported that real-life experience of OHCA would help them effectively use an AED in future responses.

### Psychological capability

Participants were knowledgeable about public access AEDs, likely locations and that the app displayed them:

> I'm aware that tube stations, Pret, the usual kind of suspects will have them, so there's quite a few within close proximity, not on my route, but if I needed help I could direct someone, go to just up the street, go to Boots. (#10)

| Facilitators |
| --- |
| Belief in the GoodSAM project and its benefits to patients |
| Debrief and follow-up after an alert |
| Knowledge that other first-responders were responding |
| Belief in the importance of AED use to patient survival |
| Planning ahead: finding AED locations |
| Confidence in AED use: helped by previous training and experience |
| Sense of professional duty |

| Barriers |
| --- |
| Concerns about managing scene |
| Concerns about managing non-cardiac arrest patients |
| Duty of care concerns |
| Less confidence in abilities |
| Reduced motivation because of experience during recent alert(s): |
| Too far from incident |
| Belief that ambulance service would arrive before them |
| Arriving after the ambulance service, even if accepting alert promptly |
| Being less likely to respond overnight |
| Prioritising arrival of scene and starting CPR as soon as possible over AED use |
| Not knowing if someone else was providing CPR at scene |
| Perceived difficulties in negotiation for AED release from its owner / custodian |
| Uncertainty about correct strategy: retrieve AED first or go direct to patient? |
| Difficulty explaining need to use AED to other bystanders |
| Concerns about acting outside normal sphere of work (healthcare professionals) |

**Figure 3** Possible barriers and facilitators to AED use: motivation. AED, automated external defibrillator.

Three participants reported knowledge of AEDs on the app that were unavailable out of hours (#12,#14 and #24). Six participants knew about an AED's location only after browsing the app (ie, when *not* receiving an alert). Four participants said they were less knowledgeable about AED locations outside their home area:

> Most places I wouldn't [know where an AED was] unless I looked on the GoodSAM app and it was on the map. (#20)

Nine reported not remembering whether the app displayed a nearby AED at the time of the alert; seven others reported that they remembered seeing an AED when alerted.

### Social opportunity

One participant (who had declined every alert that he had received) did not feel pressured to act by other bystanders, as the patient was somewhere else:

> It's easier to be influenced I think by a lack of confidence if it's not a real human being in front of you. (#21)

### Physical opportunity

No participant reported problems accessing the patient when they were first on-scene. However, 11 participants arrived after the ambulance service on their most recent alert. Three participants (#14 #17 and #25) reported this on earlier alerts as well.

Nine participants reported on their interaction with ambulance service personnel. Five participants (#5, #16, #22, #23 and #24) reported helping on scene:

> Once he realised the level of training that I have he was very excited about the fact that I was there. (#22)

However, three participants (#15, #17 and #23) reported occasions when the ambulance service did not accept assistance. Two participants (#14 and #23) additionally reported that ambulance personnel were cautious about letting them help, particularly without identification.

Five participants felt that a standard means of identifying oneself as a first responder would be useful. Seven participants were concerned about how they would find and negotiate with an AED's custodian for its removal to a different area:

> There's a whole rigmarole, can I have a defibrillator, I'm a GoodSAM volunteer, are these people going to know? (#17)

Participant 21 expressed a contrasting view:

> I kind of imagine if ever an organisation or a building or whatever has an AED and you ran in, you said somebody's having a cardiac arrest can I borrow your AED, I don't imagine that many people would say no. (#21)

A public access AED was taken to a patient once when considering participants' most recent alerts, by participant #3, who obtained it from their workplace. Participant #30 found an AED already attached to the patient on-scene. Three participants (#6, #13 and #16) took their own AEDs to the scene during their most recent alert. Eight participants reported that there were no public access AEDs close enough to retrieve when they accepted an alert.

Two other participants (#26 and #30) believed that finding an AED's exact location would be difficult. Participant #26 was unsure how one would retrieve an AED that was kept in a code-locked cabinet (which many are), and whether this code would be available through the app. Contrastingly, two participants (#5 and #13) believed that AEDs would not be too difficult to find:

> I think most of them are fairly easy accessible, I think they generally have to be in well-located places and easy to find. (#13)

Four participants commented on there being fewer public access AEDs available out of hours:

> They're in GP surgeries or local shops and both of the times I've responded they've been out-of-hours. (#12)

### Automatic motivation

Seven participants reported anxiety about on-scene management, including dealing with patients who have not suffered a cardiac arrest:

> I suppose if there was like mental health issues, or drugs and alcohol involved, I would be a bit apprehensive like getting involved in that. (#6)

Three participants (#8, #14 and #23) expressed concern about what had happened to a patient when they did not accept an alert:

> I did have that real nagging feeling afterwards about 'oh my god, you know, that person could've died because I didn't go. (#14)

Two participants (#6 and #28) said that they would welcome incident feedback or a 'debrief' from the ambulance service (#9).

### Reflective motivation

Nine participants said that GoodSAM was an important initiative. Three (#6, #9 and #16) expressed the belief that it has already saved lives.

Several factors affected or would affect the intention to accept an alert. Eleven participants cited distance to the incident. Six participants reported a belief that ambulance personnel would arrive before them, reinforced by previous experience:

> I find that very frustrating, because you drop what you're doing, go and assist someone and then by the time you get there there's already enough people so you just kind of not go. (#14)

One participant (#30) was informed by the app about other GoodSAM first responders accepting or declining the same alert but did not believe this had affected the decision to respond. Twenty participants stated a preference for going directly to the patient to assess the situation and provide CPR, rather than retrieving an AED first:

> [An AED] obviously makes a massive difference to early survival, but I think I would deem somebody doing good-quality chest compressions as a higher priority than taking an extra five to ten minutes to find the local machine. (#26)

Gaining access to an AED from its custodian might also affect the motivation to retrieve one:

> I wouldn't waste too much time… If I couldn't get it immediately within 5 to 10s I would be on my way without it. (#19)

Seven participants commented on the importance of early AED use to survival, and three (#10, #18 and #20) said they would look for an AED first if possible:

> That's the best way to save their life really. It would be worth the extra minute to get the defibrillator first. (#18)

Three participants (#13, #17 and #21) talked about the uncertainty about whether to retrieve an AED first:

> I appreciate obviously getting an early shock as quick as possible is preferable for the patient but equally I would feel bad if there was someone not doing any compressions and I had to spend five min trying to find a defib. (#13)

Thirteen participants reported that an AED's proximity to the patient or the route taken to the scene would affect their intention to retrieve it. Five participants said they had prepared by finding AED locations on the GoodSAM app before receiving an alert.

Seventeen participants explicitly expressed confidence in AED use. Reasons given were previous training (#15 and #18), previous experience (#3, #6, #10 and #11) and perceived ease of use (#11, #15 and #18). On-scene issues could affect AED use:

> Yeah I would be able to do that [use an AED], I think 90%. My hesitancy would be about dealing with the patient and the people around them. To suddenly say right we've got to get this top off and I need to connect these things straight away… So, let's say 80% confident. (#9)

Three participants (#22, #24 and #28) remarked that their sense of professional duty affected their motivation to respond to a GoodSAM alert:

> It's somewhat a large portion of what I do and who I am, so there's not many situations where I wouldn't want to respond to. (#24)

However, three responders (#5, #22 and #23—all healthcare professionals) stated concern about intervening outside their usual environment:

> It was a little bit more nerve-wracking because you walk in and you're like I want to do this, this and this but actually I can only do CPR. (#5)

### Developing interventions to improve AED use by GoodSAM first responders

We identified 10 behavioural determinants that needed to change to increase AED use, 13 potential interventions to achieve this, the behavioural change techniques required to implement them and how to deliver them (table 1).

Potential interventions were: delivering digital CPR/AED training; providing reminders about nearby AED locations; highlighting the location of the nearest AED at the time of an alert; providing access codes to AEDs in locked cabinets; providing standardised information to show AED custodians when negotiating for an AED's release; streamlining incident location and travel route information provided at the time of an alert; equipping GoodSAM first responders with their own AEDs; providing reminders about the appropriate use of AEDs; providing distance and time estimates, if travelling to the patient via the nearest AED, at the time of the alert; sending some responders directly to the patient and some via a nearby AED; providing time-to-patient (for first responder) and ambulance response time estimates at the time of the alert; delivering motivational messages related to AED use; and offering voluntary debrief after an alert.

We have also produced an infographic (online supplementary file) explaining the study and its main findings.

### DISCUSSION
### Principal findings

We identified 13 interventions that might increase AED use. Two are primarily concerned with increasing capability, five with increasing opportunity and six with increasing motivation. Some of these interventions also concern means of increasing acceptance of alerts. The two are inexorably linked: there can be no AED use if the first responder has declined the alert.

Our findings suggest a potential to improve AED use during a GoodSAM response, and these may be relevant to other volunteer first responder systems. In the interview study, one GoodSAM first responder retrieved a public access AED after an alert. In almost all cases, participants reported a capability and motivation to provide CPR and use an AED. While knowledgeable about PAD, they saw location and accessibility (particularly out of hours) of public access AEDs as barriers to use. There was concern about the time required to retrieve an AED and not knowing if bystanders were performing CPR during this time. Participants also used the app to familiarise themselves with location and access hours for public access AEDs. Many reached the patient after the ambulance

service, and this could affect motivation to respond to future alerts.

### Comparison with the literature

We have presented a number of novel findings. In addition, this study support previous findings about the lack of appropriately placed AEDs, particularly out of hours.[3] Our participants reported about not arriving first on scene and/or not intervening in an OHCA, in similarity with reports from other volunteer first responder systems.[7 10 11] Volunteer first responders in a Dutch system also reported stress, although this had resolved (in 81%) or mild (19%) by 4–6 weeks.[22]

### Implications for clinicians and policymakers

This research suggests that even capable and motivated volunteer first responders rarely use public access AEDs. We can reduce physical restraints to AED retrieval by means such as highlighting the location of AEDs, providing access codes to locked cabinets and displaying route distance and time estimates to the nearest AED and to the patient. However, a key concern about diverting to the nearest AED may always be that this increases the length of time that an OHCA patient does not receive CPR. The decision about the appropriateness of such a strategy remains that of the first responder at the time of the alert.

A key goal when establishing GoodSAM in London was the provision of early CPR, rather than a specific focus on diverting GoodSAM first responders to retrieve an AED. Patients will not always be in cardiac arrest (4 of 11 were in this study, where known) and responders will not always arrive first. This would represent a large number of occasions when an AED was retrieved for no benefit. There was a non-intentional delay of 1–4 min in activating GoodSAM in London after a call handler established a potential OHCA, which became apparent and was rectified only after completion of this study. Reducing the time between the emergency call receipt and first responder activation, and other strategies such as providing information of expected arrival times of the statutory ambulance service, may minimise the chances of a responder's assistance not being required once they reach the scene.

Early CPR is vital, so any intervention to improve AED use must not unduly delay CPR. This is possible where AEDs are easily available or if more than one person responds, with one going direct to the patient and another collecting an AED. This is likely to require more volunteers registered with the platform. AEDs can also be dispatched to the scene using taxi drivers[23] or unmanned aerial vehicles.[24 25] It is important to continue to strengthen data capture via the GoodSAM app to accurately record which interventions first responders perform during an alert.

### Strengths and weaknesses of the study

To our knowledge, this is the first study reporting a theoretically informed analysis of behaviours affecting AED use

**Table 1** Potential interventions to increase AED use in GoodSAM first responders

| What needs to change (behavioural determinants) | Potential intervention | Intervention function | Policy category | Potential behavioural change techniques | Mode of delivery |
|---|---|---|---|---|---|
| Have the physical skills to use an AED effectively (capability). | Deliver digital CPR/AED training. | Training. | Service provision: provide a specific training resource. Regulation: GoodSAM could require completion of training for continued registration with app. | Demonstration of behaviour. Feedback on behaviour. Behavioural practice/ rehearsal. | Online or in-app training (eg, interactive video-training package 'Lifesaver' (https://life-saver.org.uk). |
| Know about the existence of public access AEDs (capability). | Provide reminders about nearby AED locations. | Education. | Communication/marketing: provide regular reminders about AED locations. Guidelines: introduce recommendations about checking for AED locations into GoodSAM code of conduct. | Prompts/cues. | Information via email. Visual prompt in-app. Audio/voice prompt in-app. *Delivered at regular, spaced intervals—not at the time of an alert.* |
| Be aware of nearby public access AED locations (opportunity). | Highlight the location of the nearest AED at the time of the alert. | Enablement. | Environmental/social planning: designing the in-app environment to enable people to recognise the location of the nearest AED. | Prompts/cues. | Visual prompt in-app. Audio/voice prompt in-app. *Delivered during an alert.* |
| Be able to locate and retrieve an AED that is close enough and accessible (opportunity). | Provide access codes to AEDs in locked cabinets. | Environmental restructuring. | Guidelines: recommendations about how to access public access AEDs. Regulation: voluntary agreement with ambulance services and other AED custodians to provide access codes. Legislation: national-level laws to facilitate access to locked cabinets, for example, mandated provision of access codes. | Prompts/cues. | Visual prompt in-app. Audio/voice prompt in-app. *Delivered during an alert, specific to relevant and local AEDs only.* |
| Be able to locate and retrieve an AED that is close enough and accessible (opportunity). | Provide standardised information to show custodians of public access AEDs. | Enablement. | Guidelines: provide a document to GoodSAM first responders with recommended form of words. Communication/marketing: information campaign targeting AED custodians. | Information about health consequences. Salience of consequences. Information about others' approval. (Use of a) Credible source. | Print media campaign. Digital media campaign. Printed card (to leave with custodian). Visual display in-app (to show custodian). |
| Arrive at the patient before the ambulance service (opportunity). | Streamline location and travel route information at the time of an alert. | Enablement. | Communication/marketing: deliver 'real-time' information to GoodSAM responder via the app. Environmental/social planning: designing in-app environment to enable recognition of AED location and incident. | Prompts/cues. | Visual prompt in-app. Audio / voice prompt in-app. *Delivered during an alert.* |

Continued

**Table 1** Continued

| What needs to change (behavioural determinants) | Potential intervention | Intervention function | Policy category | Potential behavioural change techniques | Mode of delivery |
|---|---|---|---|---|---|
| Have access to one's own AED (opportunity). | Equip first responders with their own AEDs. | Enablement. | Regulation: rules designating which GoodSAM responders should be equipped with an AED and how they should respond with it. Guidelines: how and when to carry and respond with AED. | Adding objects to environment. Action planning. Review behaviour goal. Review outcome goal. | Information about appropriate use of AED delivered with equipment supplementary information via email or via the app. |
| Believe that it is appropriate to retrieve an AED (motivation). | Provide reminders about appropriate use of public access AEDs. | Education. | Communication/marketing: reminders re: appropriate AED use. Guidelines: recommendations about appropriate situations for AED use. | Information about health consequences. Information about social and environmental consequences. Salience of consequences. | Information via email. Information via app. *Delivered at regular, spaced intervals—not at the time of an alert.* |
| Believe that it is appropriate to retrieve an AED (motivation). | Provide information about route and time to the nearest public-access AED(s). | Persuasion. | Communication/marketing: deliver 'real-time' information to GoodSAM responder via app Environmental/social planning: designing in-app environment to display AED and incident location. | Information about social and environmental consequences | Visual prompt in-app. Audio/voice prompt in-app. |
| Believe that it is appropriate to retrieve an AED (motivation). | Send some responders (if multiple responders available) to retrieve a public-access AED first. | Persuasion. | Communication/marketing: sharing information about each responders' actions (so all involved know that someone is retrieving AED and someone is going to scene). Guidelines: explaining why GoodSAM responder(s) should retrieve AED first when asked. Regulation: rules on when GoodSAM responder(s) should retrieve AED first. | Information about social and environmental consequences. | Website code of conduct. E-mail updates. *Delivered at regular, spaced intervals—not at the time of an alert.* Visual prompt in-app. Audio/voice prompt in-app. *Delivered during an alert.* |
| Believe that one can retrieve an AED and reach patient before the ambulance service (motivation). | Provide time-to-patient and ambulance response-time estimates at the time of an alert. | Persuasion. | Communication/marketing: deliver 'real-time' information to GoodSAM responder via app. Regulation: organisational rules about how to safely obtain this information from ambulance services and provide it to GoodSAM responders. | Information about social and environmental consequences. Salience of consequences. Credible source. | Visual prompt in-app. Audio/voice prompt in-app. *Delivered during an alert.* |
| Be confident to use an AED effectively (motivation). | Deliver motivational messages regarding AED use. | Persuasion. | Communication/marketing: deliver motivational information directly to responder at time of alert. | (Verbal) persuasion about capability. | Visual prompt in-app. Audio/voice prompt in-app. *Delivered during an alert.* |

Continued

| Table 1 | Continued | | | | |
|---|---|---|---|---|---|
| What needs to change (behavioural determinants) | Potential intervention | Intervention function | Policy category | Potential behavioural change techniques | Mode of delivery |
| Overcome anxiety and stress about responding or using an AED (motivation). | Offer voluntary debrief after an alert (to increase AED use and/or future response rate). | Enablement. | Service provision: survey GoodSAM responders, identify those at risk of psychological harm and offer appropriate follow-up for those who need it. | Social support (emotional). Review behaviour goals. Review outcome goals. | In-app survey postalert. Face-to-face/telephone follow-up if needed. |

AED, automated external defibrillator.

in a volunteer first responder system. We have additionally developed potential interventions for testing that all those who operate similar first responder systems worldwide can consider. As a public access AED was retrieved on only one occasion, we lack information about the experience of deciding to retrieve and then use a public access AED, but this provides justification to develop and test interventions to increase AED use.

Participants occasionally reported technical problems with the app that did not fit into a TDF domain. This means a small amount of information was not coded using the TDF and was not categorised into the COM-B model. However, we shared these findings with GoodSAM for quality improvement purposes, as they impact first responders' opportunities to respond effectively.

Interviews were short, but we focused on one particular action (decision to retrieve an AED) in a group with the same exposure (a GoodSAM alert). Our response rate was low (30/248), so the sample may represent more confident and motivated first responders. We introduced targeted selection during our second recruitment window because we were oversampling those who accepted their latest alert. This will have placed a bigger value on reasons for declining alerts and reduced opportunities to identify important facilitators.

However, our recruitment method left us unable to explore the characteristics of those who did not reply to the invitation email. With a sensitive topic, participants may have been more likely to tell the interviewer what he wanted to hear.[26] CMS is a clinician and doctoral research student but did not offer clinical advice or feedback. However, this could affect how participants responded[27] or how participants interpreted information imparted at interview.[28]

### Future research

Delivering many of the proposed interventions requires a sufficient number and density of first responders. We will investigate with stakeholder groups means of doing this, including improved recruitment strategies and if the need for prior CPR certification is an absolute for registration with the app. Not all volunteer first responder systems require this. Indeed, it varies between different ambulance services using GoodSAM—in New Zealand, prior CPR certification is not required to register. Research priority exercises with GoodSAM, local ambulance services and service users will help us prioritise those interventions that could be tested before an increase in responder numbers, such as highlighting AED location and providing access codes to locked cabinets. These allow existing GoodSAM first responders to make an informed decision about whether to retrieve an AED.

The focus of this study was AED use. However, we identified that enabling any intervention before an ambulance arrived and declining an alert were important barriers to overcome to maximise GoodSAM's impact. Although some of our proposed interventions (table 1) address

these issues, overcoming these barriers may merit further research.

## CONCLUSION

A first responder used a public access AED on one occasion in this study. An AED was already attached to the patient on another occasion, and three others took their own AEDs to the scene. Despite a capability and motivation to use an AED, interview participants perceived a lack of opportunity to do so. Most believed going to the patient first to assess CPR provision was more beneficial to the patient than diverting to retrieve an AED first. Stakeholders may share this view, but there may still be the potential to improve AED use without compromising CPR provision, particularly if there are multiple responders available for each alert.

**Author affiliations**
¹Warwick Clinical Trials Unit, University of Warwick, Coventry, UK
²Warwick Medical School, University of Warwick, Coventry, UK
³Clinical Audit and Research Unit, London Ambulance Service NHS Trust, London, UK
⁴Warwick Business School, University of Warwick, Coventry, UK

**Acknowledgements** Mark Wilson, cofounder and medical director of GoodSAM, for access to GoodSAM and information relevant to this article and CMS's PhD, Deepti Bal, operations and projects director at GoodSAM, for assistance in identifying and contacting GoodSAM first responders; and members of CMS's PhD advisory group—Claire Hawkes, Christopher Hartley-Sharpe, Julian Hague (Patient and Public Involvement (PPI) representative), John Long (PPI representative)—who provided guidance and review on the project protocol, topic guides and participant information sheets.

**Contributors** CMS, FG and GDP developed the study protocol and design of participant information sheets, consent forms and topic guide. IV and RTF subsequently reviewed and edited. GDP provided guidance about clinical relevance of interview questions, while FG provided guidance about framing questions and interview technique. RTF provided specific guidance about the integration and intended use of the GoodSAM app with London Ambulance Service. CMS recruited, conducted interviews and transcribed. CMS and FG analysed interview data. FG and IV provided expert guidance regarding the application of the Behaviour Change Wheel and development of interventions. CMS drafted the manuscript for both original submission and resubmission, which all authors reviewed and edited.

**Funding** CMS is funded by a National Institute for Health Research (NIHR) Doctoral Research Fellowship (DRF-2017-10-095) for this research project. This publication presents independent research funded by the National Institute for Health Research (NIHR). The views expressed are those of the author(s) and not necessarily those of the NHS, the NIHR or the Department of Health and Social Care.

**Competing interests** CMS is an NIHR Doctoral Research Fellow, and this work was part of his PhD. He also has a volunteer role at the Resuscitation Council UK. GDP is a NIHR Senior Investigator and is supported by research grants from the NIHR, Resuscitation Council UK and the British Heart Foundation.

**Patient consent for publication** Not required.

**Ethics approval** The study received approval from University of Warwick Biomedical Sciences Research Ethics Committee on 16/03/2018 (REGO-2018–2164).

**Provenance and peer review** Not commissioned; externally peer reviewed.

**Data availability statement** All data relevant to the study are included in the article or uploaded as supplementary information. Transcripts and audio recordings are stored securely in encrypted computer files and are available to the study team if needed.

**ORCID iDs**
Christopher M Smith http://orcid.org/0000-0002-2289-8750
Frances Griffiths http://orcid.org/0000-0002-4173-1438
Rachael T Fothergill http://orcid.org/0000-0003-1341-6200

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
