## [Reviewer comments · BMJ Open]

ARTICLE DETAILS

TITLE (PROVISIONAL)	Identifying and overcoming barriers to Automated External Defibrillator use by GoodSAM volunteer first-responders in out-of-hospital cardiac arrest using the Theoretical Domains Framework and Behaviour Change Wheel. A qualitative study.
AUTHORS	Smith, Christopher Matthew; Griffiths, Frances; Fothergill, Rachael; Vlaev, Ivo; Perkins, Gavin

VERSION 1 – REVIEW

REVIEWER	Tomas Barry University College Dublin I have published in the area of community first responders for cardiac arrest, lead the Cochrane review 'Community first responders for out-of-hospital cardiac arrest in adults and children' and am a GP and 'first responder'
REVIEW RETURNED	03-Nov-2019

GENERAL COMMENTS	Thank you for the opportunity to review this interesting and important paper. Although smart tech first responder systems are increasingly being employed in OHCA care - we have little data to inform our understanding. Some comments below for your consideration - 1. You may wish to consider including GoodSAM in the title as this study relates specifically to this innovation. 2.a It would be useful (if possible) to include further key quantitative data beyond the proportion of alerts accepted to frame the issue and initiative. For instance - Overall number of presumed OHCA in London over the study period Overall number of GoodSAM alerts (and proportion that become confirmed OHCA) Proportion accepted (this is included as a %) Proportion that arrive Proportion that arrive before ambulance services Proportion that initiate CPR or attach a defib Proportion that deliver a shock 2b. It might also be useful to include some more info on the GoodSAM process/alert system - both in terms of during an alert and at the time of sign up as a volunteer. Are volunteers advised to travel directly to scene? Are they given any instructions around the integrated maps with AED locations. Is it left completely to their
---

	own devices whether they choose to detour to obtain an AED en route? 3. It should be possible to provide more specific descriptive stats for the length of interviews, range, median. 15mins suggests relatively brief interactions - Could this have been because participants had relatively little prior experience of GoodSAM/OHCA response to draw from? Do the overall trends seen in the project suggest a large population of responders with infrequent responses or a small population of frequent responders? 4. Of the 30 participants had 12 or 14 accepted their most recent alert? lines 19-30 pg 12. 11/12 arrived after the ambulance service - if this proportion reflects the overall nature of responses it will present a most significant challenge. 5. In terms of limitations, I think the uptake (30 responses / 248) means there is a risk that the data obtained might not reflect the wider experience and views of GoodSAM responders. Are participant demographic detail / experience of GoodSAM response available to help readers gain a better understanding of the participant sample ? I note three participants are identified as healthcare professionals . 6. The issue of 'accepting or declining' an alert appears to have been a major issue explored and one with significant overall importance in terms of the potential impact of GoodSAM. Does it deserve more consideration/discussion? Congrats on an important piece of work.
--	--

REVIEWER	Adjunct Associate Professor Dominic Morgan ASM University of Technology Sydney, Faculty of Health
REVIEW RETURNED	26-Dec-2019

GENERAL COMMENTS	This is an interesting paper that reads well and has a nice flow to it for the reader. It is definitely worth pursuing the suggestions to strengthen and I look forward to it being published. A few minor typos etc and other minor comments are offered for consideration below Page 5 line 17 after the word used insert "in". Page 5 line 54 after the word survive clarify the outcome measure ie survival to hospital discharge or 30 days or 1 year etc. Page 6 line 24-31. The link to the section on psychological impacts to this research could be clearer. Whilst Zjisltra's work is certainly relevant to the topic generally in this context it should be clarified whether the authors are arguing the possible impacts on wellbeing is a potential barrier to some responders attending an OHCA? If this dint come up through the research as a barrier or enabler probably could be removed. Page 8 line 26-28, for clarity just state whether the GoodSam responder figure of 19% is the same geographic area as the study. Page 8 Line 54, it refers to the SSI being analysed using the TDF. Was it
---

	“analysed” or was the TDF used to determine the structure of the questions? I understood the COM-B was used for the analysis of the outcomes of the TDF. This could be clarified. It would also strengthen your argument I think for why you used both. Page 18 Line 10 refers to “negotiating” this does not appear supported by the quote. Perhaps the authors mean “Gaining”. Page 24 line 10 after “collecting” the “and” should be “an”. Page 26 formatting error on last line Page 31 Table needs a label ie COM-B Outputs/ Analysis
--	---

VERSION 1 – AUTHOR RESPONSE

Reviewer: 1. Reviewer Name: Tomas Barry

Thank you for the opportunity to review this interesting and important paper. Although smart tech first responder systems are increasingly being employed in OHCA care - we have little data to inform our understanding.

Response: Thank you.

Some comments below for your consideration –

1. You may wish to consider including GoodSAM in the title as this study relates specifically to this innovation.

Response: The title has been amended:

“Identifying and overcoming barriers to Automated External Defibrillator use by GoodSAM volunteer first-responders in out-of-hospital cardiac arrest using the Theoretical Domains Framework and Behaviour Change Wheel. A qualitative study.”

2a It would be useful (if possible) to include further key quantitative data beyond the proportion of alerts accepted to frame the issue and initiative.

For instance –

Overall number of presumed OHCA in London over the study period.

Overall number of GoodSAM alerts (and proportion that become confirmed OHCA)

Proportion accepted (this is included as a %)

Proportion that arrive

Proportion that arrive before ambulance services

Proportion that initiate CPR or attach a defib

Proportion that deliver a shock

Response: We have addressed these issues where we can. Some of the data (proportion that arrive, arrive before ambulance service, which actions delivered on scene) were not available at the time of the study. This research team is also currently conducting work to better collect and characterise which responders arrive before the ambulance and the exact actions carried out on scene.

We have added the following statements

To the methods (Description of GoodSAM system, second paragraph):

“There were 298 OHCA in London in July 2018, and 306 in October 2018 [reference added]. There are 3-5 GoodSAM notifications daily for OHCA.”

To the discussion (implications for clinicians and policymakers):

“It is important to continue to strengthen data capture via the GoodSAM app, to accurately record which interventions first-responders perform during an alert”

2b. It might also be useful to include some more info on the GoodSAM process/alert system - both in terms of during an alert and at the time of sign up as a volunteer. Are volunteers advised to travel directly to scene? Are they given any instructions around the integrated maps with AED locations. Is it left completely to their own devices whether they choose to detour to obtain an AED en route?

Response: We have added these sentences in the methods (description of GoodSAM system, first paragraph):

“In the UK, GoodSAM first-responders require at least in-date CPR training to register with the app.”

and

“They decide themselves whether to retrieve an AED or travel directly to scene: they receive no instruction via the app.”

3. It should be possible to provide more specific descriptive stats for the length of interviews, range, median. 15mins suggests relatively brief interactions - Could this have been because participants had relatively little prior experience of GoodSAM/ OHCA response to draw from?

Response: We have added the following statements to clarify

To the methods (interview procedure):

“Interviews lasted a median of 14:56min (range 7:41-24:01)”

To the discussion (strengths and weaknesses):

“Interviews were short, but we focussed on one particular action (decision to retrieve an AED) in a group with the same exposure (a GoodSAM alert).”

Do the overall trends seen in the project suggest a large population of responders with infrequent responses or a small population of frequent responders?

Response: We do not have the data with which to answer that last point.

4. Of the 30 participants had 12 or 14 accepted their most recent alert? lines 19-30 pg 12.

Response: 12 of the first 21 interviewees had accepted their most recent alert. After we had completed 30 interviews, 14 in total had accepted their most recent alert. We have made minor textual changes to this section to clarify.

11/12 arrived after the ambulance service - if this proportion reflects the overall nature of responses it will present a most significant challenge.

Response: 11/14, as clarified above and in the text. The delay in activating GoodSAM (discussed in 'implications for clinicians and policymakers' may account for some of this. In the same paragraph (and in Table 1) we outline some of the potential other strategies that we have developed to meet this challenge.

We have also added a sentence to the discussion (future research, see also our response to point 6 below):

“The focus of this study was AED use. However, we identified that enabling any intervention before an ambulance arrived, and declining an alert were important barriers to overcome to maximise GoodSAM’s impact. Although some of our proposed interventions (Table 1) address these issues, overcoming these barriers may merit further research.”

5. In terms of limitations, I think the uptake (30 responses / 248) means there is a risk that the data obtained might not reflect the wider experience and views of GoodSAM responders. Are participant demographic detail / experience of GoodSAM response available to help readers gain a better understanding of the participant sample? I note three participants are identified as healthcare professionals.

Response: We have added the following information:

To the results (interview study):

“Eleven first-responders had previous healthcare experience. Overall figures for all GoodSAM first-responders were not available.”

To the discussion (strengths and weaknesses)

“Our response rate was low (30/248), so the sample may represent more confident and motivated first-responders”

6. The issue of 'accepting or declining' an alert appears to have been a major issue explored and one with significant overall importance in terms of the potential impact of GoodSAM. Does it deserve more consideration/discussion?

Response: We agree. AED use was the focus of this paper, but this arose as an important issue. We have re-iterated this finding and the need for further exploration in the discussion (future research, see also our response to point 4 above):

“The focus of this study was AED use. However, we identified that enabling any intervention before an ambulance arrived and declining an alert were important barriers to overcome to maximise GoodSAM’s impact. Although some of our proposed interventions (Table 1) address these issues, overcoming these barriers may merit further research.”

We have also addressed this in our response to point 1 from reviewer 2, below.

Congrats on an important piece of work.

Tomás Barry

Reviewer 2. Adjunct Associate Professor Dominic Morgan ASM

This is an important and engaging paper in an area of research with great social and economic benefit to improved outcomes. The authors should be congratulated for pursuing this topic and the direct translational benefits of the outcomes.

Response: Thank you.

There are a number of issues, mainly methodological with the paper that would strengthen it significantly for publication.

1. Is the research question or study objective clearly defined?

As a general rule the primary and secondary research questions should be clearly articulated at the end of the introduction. The study aims as written seem to represent an outcome sort rather than the research questions.

The primary research question appears to be “What are the barriers and enablers to improving the likelihood a registered GoodSam app responder will attend an out of hospital cardiac arrest (OHCA) following an automated alert”.

The secondary research question appears to be “How can the number of automated external defibrillators being taken to an OHCA be improved following an alert from the GoodSam App”.

Response: We have made this more explicit by replacing ‘study aims’ with ‘research question’ at the end of the introduction. New text reads:

“Research Question

How can we increase the numbers of AEDs taken to an OHCA by a GoodSAM first-responder following an alert?

To answer this, we took data from an interview study and used the BCW to develop a theoretically-informed list of interventions to increase public-access AED use by GoodSAM first-responders during an alert.”

Our primary concern when designing the study was the use of AEDs; it became apparent as interviews progressed that the issue of alert acceptance and attendance were inextricably linked to this fact, but it would not be accurate to present this as an a priori study objective / research question. This was badly described in the ‘Study Design’ section and has been re-written:

“Study design

Telephone semi-structured interviews performed with GoodSAM firstresponders in London, with responses coded to TDF domains. We developed topic guides based on COM-B. The focus in first-responder interviews was on decisions to use a public-access AED.”

We have then clarified the importance of this issue that arose in our discussion (future work) section:

“The focus of this study was AED use. However, we identified that enabling any intervention before an ambulance arrived and declining an alert were important barriers to overcome to maximise GoodSAM’s impact. Although some of our proposed interventions (Table 1) address these issues, overcoming these barriers may merit further research.”

2. Is the abstract accurate, balanced and complete?

“Design” label should be replaced with “Methods” to keep consistent with the rest of the paper.

Response: Corrected.

It could be clearer in the Abstract how Capability, Opportunity, Motivation relate to the Theoretical Domains Framework as it is silent on the subsequent analysis using the COM-B. Just need a reference that the TDF was used to structure/ undertake the SSI, but the responses were subsequently analysed using the COMB to determine capability opportunity and motivation.

Response: We have updated the abstract to make this clearer:

[end of objectives] “We aimed to identify means of increasing AED use by GoodSAM first-responders.

Methods: We conducted semi-structured telephone interviews, using the Theoretical Domains Framework to identify and classify barriers to AED use.

We analysed findings using the Capability, Opportunity, Motivation, Behaviour (COM-B) model and subsequently used the Behaviour Change Wheel to develop potential interventions to improve AED use.”

We have further addressed this in answer to your queries in section 3 below.

In the Results section of the Abstract page 3 at line 38 after interviews it could be strengthened by specifying who the target participant group was. Insert something like of registered GoodSam app responders who had been notified of an OHCA.

Response: We feel this is addressed in “participants”. We have amended slightly to improve clarity:

“Participants: GoodSAM first-responders alerted in the previous seven days about a cardiac arrest.”

Page 3 line6 after capability insert Opportunity which appears to have been a minor omission error.

Response: We are unable to locate this error from the description you have given.

There is a comprehensive list of findings of barriers and enablers in the paper. Only two findings are highlighted in abstract conclusions. A statement along the lines of the two listed are identified by the research as being the most important conclusions (if they are) would be appropriate.

Response: We have reworded the relevant sentences to make this clearer:

“Of particular clinical importance are: highlighting AED location; and providing route/time estimates to the patient via the nearest AED. This would assist single responders make appropriate decisions about AED retrieval.”

3. Is the study design appropriate to answer the research question?

Page 7, lines 2-30 Currently the paper does not articulate WHY the research commenced using the TDF but following interviews the outputs were then reapplied to a different Framework the COM-B before using the BCW. This would be a less common approach and would benefit from context as to why the researchers felt this was an important step to take to improve the strength of the research, as there are methodological risks with this approach.

Response: We have tried to explain the rationale between linking the TDF, COM-B and BCW sequentially (introduction, 'Theoretical frameworks to develop interventions for increased PAD use' section). We have added a more explicit explanation in the methods (Developing interventions section):

"As described {Michie, 2014 #693} we grouped and integrated TDF domains into the COM-B framework which, in turn, were linked to the BCW. Using the BCW, we identified..."

and

"Ultimately, we decided that presenting our findings – initially coded to TDF domains – according to whether they represented capability, opportunity or motivation to change behaviour, was the most accessible way to communicate our findings."

Page 9 line 33 refers to no further emails to non responders (people who didn't go to an alert), should this be non respondents? (to the email)

Response: Agreed. Corrected.

The author identifies that not all information directly fitted the TDF (Page 10 line 28-30 and the authors were alert to this but do not articulate how this was accounted for when applying the COM-B subsequently. In short the COM-B may have been applied without all of the outputs from the participants as it was excluded through the TDF. It is relevant to address how this was handled by the research team.

Response: We have clarified. There were occasional technical issues with the app that participants reported during the interviews, and this did not fit into the TDF domains. They require 'quality improvement' processes rather than de novo research to address. (Indeed, the first paragraph of the 'Physical Opportunity' section of the results concerns this and, for this reason, we have removed this from the report.) We have added a description to the strengths and weaknesses section of the discussion:

"Participants occasionally reported technical problems with the app that did not fit into a TDF domain. This means a small amount of information was not coded using the TDF and was not categorised into the COM-B model. However, we shared these findings with GoodSAM for quality-improvement purposes, as they impact first-responders' opportunities to respond effectively"

Page 12 line 2-24 The issue of potential selection bias requires some further commentary. The researchers describe randomly selecting participants in July and then targeting selection in October. Whilst the authors acknowledge this, there is a premise that the reasons a responder didn't go is more relevant than the reasons that facilitated the desired behaviour ie it is equally relevant that a responder acted as desired because they had flexible work arrangements and could leave immediately, as a person who didn't go saying they didn't have flexible work arrangements and therefore couldn't go. This has the potential to exclude a series of enablers and as such the mixed selection may be problematic to replicating the study. Some statements dealing with how the issue was addressed is important and acknowledging that it could be a potential weakness.

RESPONSE: We acknowledge the weakness in our approach as you have described. We have added an explanation in the Strengths and Weaknesses section:

“We introduced targeted selection during our second recruitment window because we were over-sampling those who accepted their latest alert. This will have placed a bigger value on reasons for declining alerts and reduced opportunities to identify important facilitators.”

4. Are the methods described sufficiently to allow the study to be repeated? Page 20 line 15 – 19. At this stage the research outcomes could not be reproduced due to the reference to “drawing upon our knowledge of the existing literature as well as findings from the interview study”. This would suggest that this is no longer an original research paper, but a systematic review of other research by other authors combined with original research. This does need to be addressed prior to publication. In reality either all findings from other research need to be individually identified and cited correctly (in both the text and tables) or be removed and rely solely on the outcomes of this research. This would be the stronger methodological approach.

Response: Apologies, this is the fault of the first author. This work forms part of his PhD that included a systematic review, and in the PhD there is a wider discussion comparing the original research findings and our prior knowledge. Everything reported in part 1 of the results is an original finding from this study, and the intervention development (part 2) is based on solely original findings from this research study. The offending statement had previously been edited out but crept back in to the latest version of the manuscript. It has been removed.

5. Are research ethics (e.g. participant consent, ethics approval) addressed appropriately?

Yes

6. Are the outcomes clearly defined?

In terms of presentation yes, but primarily for the reasons above ie it cant be identified what are the outcomes of this research compared to systematic review it is not currently clear.

Response: We have addressed this in section 4 above.

7. If statistics are used are they appropriate and described fully?

N/A

8. Are the references up-to-date and appropriate?

Individual references have not been checked for mapping to text, however all appear contemporary and relevant from the list provided

9. Do the results address the research question or objective?

The research questions would benefit by being more clearly articulated as per section 1 and then the results should be re-written removing reference to results identified in other studies, as per section 4

Response: As indicated, we have addressed these points in sections 1 and 4 above.

10. Are they presented clearly?

Visually appropriate and elements of discussion appear in results as listed

11. Are the discussion and conclusions justified by the results

The discussion covers the topic well and the benefits of the findings. Once the conclusions are addressed this section will be fine.

Response: Thank you. Hopefully we have addressed all of these points above.

12. Are the study limitations discussed adequately?

It would be more common to show Strengths and Limitations at the end of the discussion, rather than following Introduction. Some minor areas of weakness previously discussed and the issues of potential study bias will see this completed fairly easily.

Response: We have added to the section, as discussed above, and have moved its position as suggested.

13. Is the supplementary reporting complete (e.g. trial registration; funding details; CONSORT, STROBE or PRISMA checklist)?

No supplementary reporting

Response: To note, we completed and attached an SRQR checklist (both to the original submission and to this re-submission)

14. To the best of your knowledge is the paper free from concerns over publication ethics (e.g. plagiarism, redundant publication, undeclared conflicts of interest)?

15. Is the standard of written English acceptable for publication?

This is an interesting paper that reads well and has a nice flow to it for the reader. It is definitely worth pursuing the suggestions to strengthen and I look forward to it being published.

A few minor typos etc and other minor comments are offered for consideration below

Page 5 line 17 after the word used insert "in".

Response: Corrected.

Page 5 line 54 after the word survive clarify the outcome measure ie survival to hospital discharge or 30 days or 1 year etc.

Response: Added "to hospital discharge"

Page 6 line 24-31. The link to the section on psychological impacts to this research could be clearer. Whilst Zijlstra's work is certainly relevant to the topic generally in this context it should be clarified whether the authors are arguing the possible impacts on wellbeing is a potential barrier to some responders attending an OHCA? If this didn't come up through the research as a barrier or enabler probably could be removed.

Response. We have removed. Anxiety and stress are mentioned in the results, and an intervention proposed in Table 1. We agree that the original paragraph in the introduction was presented with little context, and this change should improve this. We already referred to the Zijlstra paper in the discussion and have now expanded on the discussion there instead.

"Volunteer first-responders in a Dutch system also reported stress, although this had resolved (in 81%) or mild (19%) by four to six weeks {Zijlstra, 2015 #988}."

Page 8 line 26-28, for clarity just state whether the GoodSam responder figure of 19% is the same geographic area as the study.

Response: Added "in London"

Page 8 Line 54, it refers to the SSI being analysed using the TDF. Was it “analysed” or was the TDF used to determine the structure of the questions?
 I understood the COM-B was used for the analysis of the outcomes of the TDF. This could be clarified. It would also strengthen your argument I think for why you used both.

Response: The phrase “analysed using the TDF” has been replaced by “with responses coded to TDF domains”. The wider concerns have been addressed in section 3 above.

Page 18 Line 10 refers to “negotiating” this does not appear supported by the quote. Perhaps the authors mean “Gaining”.

Response: “Negotiating access to” replaced by “Gaining access to”

Page 24 line 10 after “collecting” the “and” should be “an”.

Response: Corrected.

Page 26 formatting error on last line

Response: Corrected.

Page 31 Table needs a label ie COM-B Outputs/ Analysis

Response: Added.

VERSION 2 – REVIEW

REVIEWER	Tomás Barry University College Dublin, Ireland I am a GP with roles in cardiac arrest education, research, and clinical care.
REVIEW RETURNED	21-Feb-2020

GENERAL COMMENTS	Thanks for the opportunity to review this updated version which will make a valuable contribution to the literature in this area. My one comment final comment would be whether to remove 'A first-responder used a public-access AED on one occasion in this study. An AED was already attached to the patient on another occasion, and three others took their own AEDs to the scene' from the conclusion. This study is primarily about gaining a more in-depth understanding of the potentials to increase GoodSAM AED use and the authors parallel observational quantitative work will represent the more valid means of establishing how frequently GoodSAM activation's result in AED use at population level.
---

REVIEWER	Adjunct Associate Professor Dominic Morgan ASM University of Technology, Sydney
REVIEW RETURNED	15-Feb-2020

GENERAL COMMENTS

Well done on reporting this interesting and important piece of research. The issues around research questions and methodology have been addressed and where judgements have been made about the approach (which may have led to bias) have been articulated as to their basis and management. I look forward to seeing it published in due course.